# Effect of Basalt/Steel Individual and Hybrid Fiber on Mechanical Properties and Microstructure of UHPC

**DOI:** 10.3390/ma17133299

**Published:** 2024-07-04

**Authors:** Yongfan Gong, Qian Hua, Zhengguang Wu, Yahui Yu, Aihong Kang, Xiao Chen, Hu Dong

**Affiliations:** 1College of Civil Science and Engineering, Yangzhou University, Yangzhou 225100, China; yfgong@yzu.edu.cn (Y.G.); hmh197599@163.com (Q.H.); ahkang@yzu.edu.cn (A.K.); 2Jiangsu Ruiwo Construction Group Co., Ltd., Yangzhou 225600, China; yuyahui2024@163.com (Y.Y.); gycx1987@126.com (X.C.); 3Jiangsu SinoRoad Transportation Science and Technology Co., Ltd., Nanjing 211806, China; dh1034812505@163.com

**Keywords:** UHPC, basalt fiber, hybrid fibers, fracture energy, micro-structure

## Abstract

Ultra High-Performance Concrete (UHPC) is a cement-based composite material with great strength and durability. Fibers can effectively increase the ductility, strength, and fracture energy of UHPC. This work describes the impacts of individual or hybrid doping of basalt fiber (BF) and steel fiber (SF) on the mechanical properties and microstructure of UHPC. We found that under individual doping, the effect of BF on fluidity was stronger than that of SF. Moreover, the compressive, flexural, and splitting tensile strength of UHPC first increased and then decreased with increasing BF dosage. The optimal dosage of BF was 1%. At a low content of fiber, UHPC reinforced by BF demonstrated greater flexural strength than that reinforced by SF. SF significantly improved the toughness of UHPC. However, a high SF dosage did not increase the strength of UHPC and reduced the splitting tensile strength. Secondly, under hybrid doping, BF was partially substituted for SF to improve the mechanical properties of hybrid fiber UHPC. Consequently, when the BF replacement rate increased, the compressive strength of UHPC gradually decreased; on the other hand, there was an initial increase in the fracture energy, splitting tensile strength, and flexural strength. The ideal mixture was 0.5% BF + 1.5% SF. The fluidity of UHPC with 1.5% BF + 0.5% SF became the lowest with a constant total volume of 2%. The microstructure of hydration products in the hybrid fiber UHPC became denser, whereas the interface of the fiber matrix improved.

## 1. Introduction

Ultra-high performance concrete (UHPC) is a novel type of concrete material with distinct performance indices [1,2], comprising cement, mineral admixtures, admixtures, fine aggregates, fibers, etc. UHPC is widely utilized in the railroad, road, bridge, and building sectors [3]. Unlike conventional concrete, UHPC can provide ultra-high compressive strength of up to 120 MPa and reasonable bending strength [4], with high cement-use efficiency, a dense microstructure, ultra-high mechanical characteristics, and durability [5,6]. Despite providing excellent compressive strength and stiffness, UHPC has limitations, including poor toughness, weak bending, and tensile qualities [7]. The existing approach to addressing these limitations is adding fibers to the matrix to boost UHPC performance. Fiber may effectively reduce the spread of cracks and dissipate the fracture energy of the cementitious material [8]. Common fiber types include steel fiber, mineral fiber, organic fiber, and glass fiber. The right quantity of fiber is interwoven and overlapped in the concrete to create a spatial skeletal structure. The fiber can reinforce and toughen UHPC, bridging cracks and reducing them [9,10].

According to numerous research, UHPC’s toughness, flexural strength, and compressive strength can all be significantly increased by adding steel fiber (SF) [11,12]. In order to directly bridge diagonal cracks or indirectly disperse stress to both sides of the crack, steel fibers can be employed like stirrups [13]. Abba et al. [14] discovered that the compressive strength of UHPC with 3% SF can reach 164 MPa, 12 MPa higher than that with 1% SF. Jin et al. [15] noted that the tensile strength of UHPC with 3% SF can become approximately 2.4 times greater (16.05 MPa) than samples without SF (6.67 MPa). However, SF is expensive and has a high density, as well as poor corrosion resistance. Moreover, carbon dioxide emissions generated during steel fiber production exhibit adverse effects on natural resources and the environment [16].

Basalt fiber, a novel type of eco-friendly fiber with good compatibility, thermal stability, chemical stability, and corrosion resistance [17,18], can effectively control concrete contraction and cracks expansion, thereby significantly increasing the mechanical properties, bending toughness, and durability. BF is lighter with similar mechanical quality to SF. Nevertheless, basalt fibers have low energy consumption and are cheaper than steel fibers, providing additional economic benefits [19]. Chen et al. [20] used BF to reinforce UHPC and discovered that the UHPC with BF has better flowability and fiber distribution, unlike that with SF. Wu et al. [21] noted that BF exhibits a larger size, which causes a smaller specific surface area and fiber-distribution density in the UHPC slurry, resulting in a reduction of resistance for UHPC slurry flow. Jiang et al. [22] confirmed the feasibility of using BF instead of SF to strengthen ultra-high-performance seawater sea–sand concrete, geared towards resolving SF corrosion caused by chloride ions. Elsewhere, Nuaklong et al. [23] found that adding basalt fiber increased the elastic modulus of UHPC by 2.67% and improved sample transverse ductility. Due to its affordability and environmental friendliness, BF is typically utilized as a reinforcing material in a variety of areas. Its advantages include resistance to acids and alkalis, high strength, and resilience to both high and low temperatures [24,25,26].

Hybrid fiber concrete is stronger and more durable than single-fiber concrete [27]. This is because small-diameter fibers can successfully connect macrocracks, large-diameter fibers may effectively block microcracks from preventing the development of macrocracks, and hybrid fibers can exert a good synergistic impact to improve concrete performance. Meng et al. [28] ascertained that a hybrid fiber combination could effectively improve the compressive strength of UHPC. Lai et al. [29] stated that the combined use of SF and BF can promote the resistance of UHPC to projectile effects and explosions. Niu et al. [30] discovered that fibers of varying hybridized geometries and materials can improve the mechanical properties of UHPC. Li et al. [31] found that hybrid fiber-reinforced UHPC exhibits good ductility by mixing steel fibers with seven nonmetallic fibers. Lin et al. [32] analyzed the dynamic properties of UHPC enhanced by hybrid POMF-SF, indicating that hybrid fibers increase the dynamic mechanical strength of UHPC.

Now, several studies have indicated that fiber can improve the majority of UHPC characteristics. However, distribution, content, and hybrid combination of fibers influence the mechanical properties and durability of UHPC [33]. The mechanical properties of UHPC may be reduced when the fiber content is too high, resulting in the opposite effect [34,35]. Khayat et al. [36] noted a critical value of fiber content, beyond which the flowability of UHPC can be significantly minimized. Xu et al. [37] found that SF content increased the bonding surface area between the fiber and UHPC, thereby improving the tensile properties but reducing the fluidity of UHPC. Therefore, it is important to investigate the effects of doping hybrid basalt and steel fibers of UHPC. Simultaneously, research on UHPC materials mainly focuses on steel and organic fibers, with less emphasis on basalt fibers since inorganic fibers have little effect on toughness. However, the resin-twisted basalt fiber (BF) has benefits, including a strong anchoring effect and good dispersibility due to its unique ribbed surface shape. The toughening effect is superior to short-cut basalt fibers. As a result, the importance of hybrid reinforced fiber on the mechanical properties of UHPC is currently an important area of research.

This work investigated the effects of individual or hybrid BF and SF doping on the mechanical properties of UHPC. First, we investigated the effects of single-doped fibers on the functional and mechanical properties of UHPC. Secondly, the functional and mechanical characteristics of hybrid fiber UHPC were examined by partially substituting SF with BF at 25%, 50%, and 75% replacement rates. We elucidated the mechanism of fiber in the matrix in conjunction with SEM scanning examination.

## 2. Materials and Methods

### 2.1. Raw Materials

The BF used in the test was produced by Jiangsu Tianlong Basalt Continuous Fiber Co., Ltd. (Yangzhou, China), and the SF was produced by Jiangsu Subot New Materials Co., Ltd. (Nanjing, China). Figure 1 shows the appearance of the fibers. Table 1 presents the fiber-characteristic parameters [38].

### 2.2. Design Mixes

P·II 52.5 Portland cement is the cementitious material used in the test. The slag powder used in the test is S95 and quartz sand produced by Shanghai Baosheng Construction Engineering Co., Ltd. (Shanghai, China). A water reducer with a 30% water-reducing capacity was also added to the design combination to increase uniformity and processability.

The test was split into two halves, and we used blank UHPC without fiber as the control group.

Part 1: We investigated the effect of BF and SF on the mechanical characteristics of UHPC when introduced independently, with a fiber content of 0.25%, 0.5%, 0.75%, 1.0%, 1.5%, and 2.0%. Table 2 shows the proportion of single-fiber blending. The specimens with individual doping of BF were named UHPC-BF, and the specimens with an individual doping of SF were named UHPC-SF.

Part 2: Keeping the overall volume content of 2% constant, we replaced some steel fibers with basalt fibers at 25%, 50%, and 75% replacement rates. We examined how the combination of basalt fibers and steel fibers influences the mechanical characteristics of UHPC. Table 2 presents the mix ratio of hybrid fibers, with the initials BF and SF standing for basalt fibers and steel fibers, respectively. Additionally, the previous values denote the proportions of basalt and steel fibers. For instance, 05BF15SF stands for UHPC with 1.5% steel fibers and 0.5% basalt fibers.

The UHPC specimens were prepared by JJ-5 cement mortar mixer. Cementitious material (cement, mineral admixtures) and sand were put into the mixing pot and stirred for 2 min to ensure good dispersion and achieve higher fiber matrix interface performance. The superplasticizer and water were mixed evenly, and then half of them were added into the mixing pot and stirred for 6 min. The fibers were added to the mixtures lowly and stirred for 2 m to disperse evenly. Finally, the UHPC slurry was poured into the mold, and the specimen was formed by shaking table for 2 m. The formed specimen was put into the standard maintenance room for 24 h and then removed into the standard curing room until the specified age.

### 2.3. Testing Methods

#### 2.3.1. Fluidity Test

Since UHPC raw materials are all fine aggregates with small particle sizes, the UHPC slurry flow test method was referred to as the “Methods for Testing Uniformity of Concrete Admixtures” GBT8077-2023 [39]. UHPC slurry was poured into a frustum round mold and lifted to measure slurry flow. The truncated cone round die had an upper diameter of 70 mm, lower diameter of 100 mm, and height of 60 mm, with a smooth inner wall with no seams.

#### 2.3.2. Compressive Strength

A reference was made to the “Testing Method for Cement Mortar Strength (ISO Method)” GB/T17671-2021 [40] to analyze the compressive strength of UHPC. A YA-600 microcomputer-controlled fully automatic pressure testing machine was used to test the compressive strength of UHPC. Prepare three UHPC cube test blocks of 100 mm × 100 mm × 100 mm each, and take the arithmetic mean of the calculated compressive strength values of the three test blocks as the test result.

#### 2.3.3. Flexural Strength

The servo-hydraulic multifunctional material testing system UTM-25 was used to compute the flexural strength of the mortar specimens in this experiment. Prepare three UHPC specimens with a mix ratio of 100 mm × 100 mm × 400 mm for each group, and take the arithmetic mean of these specimens as the final result of the flexural strength value of the prism in that group.

#### 2.3.4. Splitting Tensile Strength

The splitting tensile strength of UHPC was evaluated based on the “Standard for Testing Methods for Physical and Mechanical Properties of Concrete” GB/T50081-2019 [41]. This test was conducted using the YA-600 microcomputer-controlled fully automatic pressure testing machine produced by Changchun Kexin Testing Instrument Co., Ltd. (Changchun, China) Prepare three UHPC cube specimens of 150 mm × 150 mm × 150 mm each and take the arithmetic mean of the calculated splitting tensile strength values of the three specimens as the test result.

#### 2.3.5. Fracture Toughness

The fracture test is the three-point bending fracture test in the “Hydraulic Concrete Fracture Test Specification” DL/T5332-2005 [42], and the concrete fracture energy specimen measures 40 mm × 40 mm ×160 mm. The effective span L of the specimen was 120 mm. The mid-span of the specimen was cut with a depth of 12 mm and a width of 2 mm. Figure 2 shows the geometric schematic diagram of the specimen. The servo-hydraulic multifunctional material testing system UTM-25 was used for the fracture-toughness test.

Fracture energy refers to the energy consumption of the fracture zone per unit area, which can be determined by the area under the load-displacement curve. The calculation is shown in Equation (1).
(1)$$G=\frac{W+mg\delta_{0}}{A}$$
where *G* is the fracture energy (J); *W* is the measured P-δ surrounding area of the curve and abscissa, *A* is the area of the fractured ligament of the specimen (mm^2^); $\delta_{0}$ is the displacement of the loading point during final failure (mm); *m* is the mass of the specimen (kg); and *g* is the acceleration of gravity.

## 3. Results

### 3.1. Effect of Basalt/Steel Individual Fiber on Properties of UHPC

#### 3.1.1. Fluidity

Figure 3 shows the effect of BF and SF on the fluidity of UHPC. As shown, the maximum flow occurred at zero fiber content, and the flow value was 310 mm. The flow gradually decreased with increasing fiber content, and more fiber made UHPC more viscous and less fluid. This is because an increase in the number of fibers per unit volume increases the difficulty of particle movement within the matrix and hinders the movement between particles, the fibers increase the friction between them and the matrix, and they are randomly distributed in the UHPC matrix acting as a skeleton, ultimately preventing the flow of UHPC. In addition, an increase in the specific surface area of the fiber increases the demand for cement slurry covering the fiber and aggregate surface, as well as decreases matrix fluidity.

With the same content, the fluidity of BF decreased more significantly than that of SF. With the fiber content increasing from 0% to 1%, the flow of UHPC-BF decreased by 80 mm (26%), whereas that of UHPC-SF decreased by 50 mm (16%). The fluidity of UHPC with 1% BF was 11.5% lower than that of UHPC with 1% SF. As shown in Figure 4, this is because the SF surface is smoother, whereas the BF surface has a ribbed shape and is rougher, increasing the friction between it and the matrix, and thereby complicating the molecular movement within the matrix. Each SF content conformed to the flow state UHPC. However, when the BF content reached 1.5%, the fluidity of UHPC-BF became lower than 200 mm, which did not meet the requirements of fluidity degree. Therefore, the addition of UHPC-BF was halted. Since the fluidity of the specimen at this time was poor, the internal porosity and structural damage increased, whereas the strength decreased.

#### 3.1.2. Compressive Strength

Figure 5a shows that BF can considerably increase the initial strength of UHPC. The compressive strength of UHPC-BF 7 d at a 0.5% dosage was 116.1 MPa, i.e., higher than that of UHPC-SF at the same dosage and up 29.7% from the control group. When UHPC was 7 days old, the compressive strength of 1% BF added reached 92.1% for 28 days, and 1.0% SF added reached 90.4% for 28 days. This is because the development of cracks is what causes the failure of UHPC; adding BF and SF can significantly reduce cracks while at the same time improving specimen strength [43].

Figure 5b shows that the overall compressive strength of UHPC-BF has a pattern in the figure that first increases and then decreases with increasing BF dosage. The highest compressive strength at this time was 130.6 MPa, 28.3% higher than the blank group. The ideal dosage was 1%. The compressive strength of UHPC-SF gradually increased with increasing dosage. Before 1% dosage, the compressive strength quickly increased but gradually increased at ~1%–2% dosage. UHPC-SF typically has a 28 d compressive strength higher than UHPC-BF. This is because the compressive strength of UHPC increases at a specific fiber content due to the fiber-bridging effect. Nonetheless, if the fiber concentration is too high, the specimen becomes less fluid and suffers from more internal structural damage, thereby lowering the compressive strength. Due to its massive strength, SF may successfully join macroscopic cracks in the middle and late stages of crack formation. UHPC hydration is completed at this point, and the strength of the matrix increases. The matrix and SF adhere better, and the compressive strength of UHPC-SF significantly increases.

#### 3.1.3. Flexural Strength

Figure 6a shows the variations in the flexural strength of UHPC with single-fiber concentration at 7 days. At 7 days, BF has a stronger improvement effect on the flexural strength of UHPC than that of SF at a dosage of 0–1%. Also, 1% UHPC-BF has a flexural strength of 26.4 MPa, which is 6.5% more than UHPC-SF. This is because BF has a rougher surface than SF and has superior adhesion with UHPC in the early stages due to more matrix friction. Cracks cause failure, and the fibers near the crack may help to carry stress. Moreover, since BF is highly dispersible, more fibers bear the force on the fracture surface, hence increasing the flexural strength of UHPC.

Figure 6b shows the changes in single-doped fiber and the flexural strength of UHPC after 28 days. As shown, the flexural strength of UHPC initially increased and then decreased with increasing BF content. The strength is currently 32.6 MPa, i.e., 34.7% stronger than the blank group, with an ideal content of 1%. UHPC steadily increases as SF content increases, reaching an ideal SF content of 2% and a strength of 36.5 MPa. UHPC-SF shows greater flexural strength than UHPC-BF at high fiber contents. This is due to the limited dispersibility of SF and, therefore, fewer fibers on the stress surface at low dosages. Secondly, although BF has a greater coefficient of friction and performs better while bonding to the matrix, the fluidity of the specimen and its flexural strength decreases, while its internal loss increases when the dosage of BF is above 1%.

With the addition of fibers, the flexural strength of UHPC was considerably enhanced, and its brittle failure mode changed to ductile failure. Figure 7 shows morphology failure from the flexural strength test of UHPC. Upon loading of fiber UHPC, microcracks are first created, which are subsequently supported by other fibers, causing the propagation of microcracks. The stressed fiber snaps and pulls out due to constant load increase, and the microcrack progressively grows into a macrocrack, hence hurting the specimen. After failure, the specimen is not split in half and is kept together by fibers to preserve integrity.

#### 3.1.4. Splitting Tensile Strength

Figure 8 compares the splitting tensile strength of UHPC with single SF and BF doping. UHPC increased by 45%, 55%, 76%, 112%, and 87.9% compared to the control group when the BF content was 0.25%, 0.50%, 0.75%, 1.0%, and 1.5%, respectively. BF has a stronger capacity to prevent fracture propagation and a better connecting effect on the matrix at a dosage of 1% [44]. The specimen currently has the highest splitting tensile strength. The splitting tensile strength of UHPC decreases at 1.5% dosage. High BF content decreases the UHPC flow performance, develops internal pores, and exacerbates structural degradation, all of which reduce splitting tensile strength. However, the dosage is inadequate, and few fibers are fractured, with a weak effect of bearing stress. The best BF dosage at this time was 1% because the splitting tensile strength was rather low. The splitting tensile strength of UHPC increased by 66.7%, 91%, 112.1%, 182%, 194%, and 118% compared to the blank group at steel fiber contents of 0.25%, 0.5%, 0.75%, 1%, 1.5%, and 2%, respectively. The optimal content of steel fiber was 1.5%, and the tensile strength of UHPC initially increased before decreasing with increasing steel fiber content. This is because when the fiber content increases, more fibers are added per unit area, increasing the amount of cement slurry surrounding the fibers and decreasing the amount of hydration products the matrix produces, hence leading to the original flaws in the UHPC matrix.

### 3.2. Effect of Hybrid Fiber on Properties of UHPC

#### 3.2.1. Fluidity

As shown in Figure 9, the fluidity of UHPC gradually decreased with an increasing BF replacement rate, and the maximum fluidity of primary UHPC was 310 mm. The flow degree of the dynamic UHPC was 200 mm, and the minimum flow value of the UHPC of 15BF05SF was 175 mm, which does not meet the requirements of the dynamic UHPC of the flow. When the BF replacement rate was below 50%, the fluidity exceeded 200 mm, indicating enhanced flow performance. However, the addition of fiber hindered the working performance of UHPC, which was attributed to the restraining and blocking effects of fibers. The fiber tended to wind and overlap, forming clusters that made the matrix sticky. The rough surface morphology of BF led to increased friction with the matrix, and its special ribbed surface created an anchorage effect, causing a more noticeable decrease in UHPC fluidity. As BF is hydrophilic, it adsorbs a small amount of water during the preparation process, reducing UHPC mobility. Hence, the replacement rate of BF should not exceed 50%. Notably, SF has a smooth surface and less friction, hence, there is less effect on the flow of UHPC than BF.

#### 3.2.2. Compressive Strength

Figure 10 shows the cubic compressive strength of UHPC at 3 d, 7 d, and 28 d for each group. The 7 d compressive strength of 10BF10SF was the highest, indicating that BF addition improves the early strength of UHPC. This is due to the rougher surface of BF, denser cement fiber interface at the early stage of hydration, and hydrophilic sizing agent on the BF surface, causing more C-S-H gel to adhere to the fiber surface and tighter bonding between fibers and matrix. Secondly, the particular twisted structure of BF has a higher matrix-anchoring effect and synergistic matrix deformation. Its compressive strength significantly increases in the early stages of hydration. However, due to its weak stiffness, it does not significantly increase in the latter stages of hydration compared to steel fibers. Its compressive strength significantly increases during the early stages of hydration. However, due to its weak stiffness, it does not significantly increase during the latter stages of hydration, unlike steel fibers. Fiber-mediated enhancement of the internal structure of UHPC, which can promote strength increase in the early stages, is responsible for the 3 d strength of hybrid fiber UHPC reaching 80% of the 28 d strength. By distributing stress via significant fissures, fibers can increase the ductility, strength, and load-bearing capacity of composite materials based on cement.

#### 3.2.3. Flexural Strength

Figure 11 shows the flexural strength of composite fiber UHPC. The flexural strength of composite fiber UHPC initially increased and then decreased with an increasing BF substitution rate. The flexural strength of UHPC was at its best upon the addition of 1.5% SF and 0.5% BF; its corresponding 7 d and 28 d flexural strengths increased by 1.49 and 1.75 times, respectively, compared to the control group. The flexural strength increased by an increasing rate of BF substitution. However, significant BF weakened the flexural strength of UHPC. It has a better anchoring effect on the substrate majorly due to the distinctive twisted structure on the BF surface. BF predominantly increased in early UHPC strength with a more restraining effect on its microcracks during the initial stage of load action. Additionally, the inclusion of BF can improve the internal structure of UHPC, and the high dispersibility of material can promote the development of a denser microstructure.

#### 3.2.4. Splitting Tensile Strength

Figure 12 illustrates that the splitting tensile strength of UHPC initially increases and declines afterwards as the BF replacement rate increases. The UHPC material exhibits the highest splitting tensile strength (05BF15SF). This is ascribed to the impact of two types of fibers on crack propagation at various points throughout the failure process. The incorporation of microfibers delays the development of macroscopic cracks compared to single-type big-fiber concrete. Moreover, BF can promote fiber dispersion and terminate the development of microcracks. BF exhibits a more rugged morphology, greater length, and enhanced bonding impact with the matrix in comparison to SF. During the initial loading stages, it exerts a more significant inhibiting effect on microcracks. Additionally, BF has stronger dispersibility compared with SF. The weight of SF makes it vulnerable to deposition during the stirring molding process, allowing for the entry of more air into the interior of UHPC, increasing UHPC flaws and decreasing the number of stress fibers at the cracking site. However, the ongoing increase in the BF replacement rate decreases the splitting tensile strength of UHPC due to the higher strength of SF, which resists the emergence of macroscopic cracks. Fiber mixing can improve the splitting tensile strength of UHPC more effectively than single fibers, demonstrating a beneficial synergistic effect between SF and BF. Fiber blending achieves this by restricting early micro-cracks and later macroscopic cracks.

### 3.3. Effect of Fracture Toughness on UHPC

#### Effect of Single-Doped/Multi-Doped Fiber on Fracture Toughness

As indicated by the load-disturbance curve in Figure 13, we infer that BF enhances the toughness of UHPC. The fracture energy of the specimen is significantly improved, measuring 102% higher than that of the control group. The fibers are prone to fracture once they are loaded, and they lack the capacity to remain connected to the cracks because of the low strength of BF itself. This is because microcracks primarily form inside the specimen during the early stress stage, and the BF surface is ribbed, creating a stronger anchoring force with the matrix. Consequently, it can improve the initiation load and inhibit microcrack formations in the specimen. Nevertheless, at a 1% SF content, the limited quantity and uneven dispersion result in reduced SF to withstand stress on the fracture surface of the UHPC matrix. The peak strength of UHPC-SF is inferior to that of UHPC-BF, primarily due to the modest anchoring effect of the UHPC matrix. However, the steel fiber itself has high strength and can effectively connect macro-cracks after UHPC fracture, inducing ductile failure in specimens.

The fracture energy of UHPC first increases and then falls as the replacement rate of BF rises. The maximum fracture energy, which is 8.1% higher than that of 2% SF, is detected at 05BF15SF. Not only does 05BF15SF surpass 2% SF in peak load, but it also exhibits a larger load-displacement envelope area. Steel fibers play a key role in imparting ductility to UHPC, and the incorporation of BF enhances both the initiation load and peak load. Composite-doped fiber UHPC specimens demonstrate superior fracture performance compared to UHPC specimens with a single dopant of steel fiber. This is because microcracks, which first develop in the weak zone, are significantly inhibited by BF in the early stages of specimen cracking. Consequently, stress builds up at the crack tip. The BF enhances the initiation load and peak load by reducing local stress concentration and inhibiting fracture propagation due to its potent anchoring effect. Through the partial substitution of BF for steel fiber, effective fiber-bridging resistance was achieved, extending up to larger crack widths and preventing the isolated crack from propagating. Furthermore, it resulted in the opening of a wider crack at the location of the greatest fiber-bridging stress. Combining BF with SF improves the internal structure of UHPC, optimizes the spatial distribution of fibers within UHPC, alleviates subpar SF cluster dispersion performance, and increases the strength of UHPC and toughness. Table 3 illustrates that the 15BF05SF minimum fracture energy is 17.5 kJ/m^3^. Because of the quick drop in flowability caused by the number of BF beyond the limit value (1%), the inhibitory impact of fibers on macroscopic fractures decreases, internal flaws in UHPC increase, and the material’s peak strength and toughness start to decline.

### 3.4. Microstructure Analysis of UHPC

#### 3.4.1. Basalt/Steel Fiber Individual of UHPC

Once fibers are incorporated, the fracture surface of the notched specimen extract SF and breaks BF, both of which are uniformly distributed. The fibers are somewhat bent after being pulled out. To more clearly demonstrate how different fibers affect the fundamental characteristics of UHPC, the microstructure of 28-day UHPC specimens was examined using the SEM. Figure 14a displays the internal microstructure of blank UHPC, showing its hydration products, unhydrated cement particles, and holes, with calcium hydroxide crystal and C-S-H gel acting as the principal hydration products. The compactness of the microstructure can also be influenced by the detrimental pores between particles. The microstructure of UHPC was much more compact after the addition of SF due to increased hydration products, tight bonding between fibers and matrix, and decreased porosity, as shown in Figure 14b. The interface between SF and UHPC exhibits some flaws, and SF is tough and difficult to break. Steel fibers may separate during the stress process owing to the flat surface of SF and its loose attachment to the substrate. In Figure 14c, a superior adhesion between BF and the substrate is evident, with more hydration products attached to the surface of BF compared to SF. This is attributed to the rougher surface of BF and the application of hydrophilic wetting agents. The dense and homogeneous microstructure around BF facilitates strong adhesion between fibers and cement slurry. The fibers can bridge the microcracks formed when cement slurry is loaded, allowing the specimen to carry the load even after it has cracked. To minimize crack extension and provide ultra-high strength for UHPC, the high adhesion between BF and cementitious materials transfers stress to fibers, transforming it into fiber tensile strength.

#### 3.4.2. Hybrid Fiber of UHPC

The stress-damaged specimens and the control specimens were examined at 28 d using SEM at various magnifications. Results showed that the resistance mechanisms of UHPC doped with SF + BF and blank UHPC were highly distinct. A well-formed plate-like calcium hydroxide crystal frame can be observed in the UHPC specimen. Although the internal microstructure of fiber-doped UHPC is more compact compared to that of blank UHPC, it contains more calcium hydroxide crystals and less hydration products. This is because fibers reduce the pores and alter the UHPC’s internal structure. Within UHPC, the fibers are more likely to create a spatial network structure, which minimizes the matrix’s pores and fosters a stronger connection between hydration products and fibers. Figure 15 illustrates the mechanism through which the hydrophilicity of BF promotes the absorption of the gel on its surface to enhance the growth of hydration, such as C-S-H gel, which fills the pores in the matrix and boosts the test piece’s strength. Additionally, fiber debonding is inhibited and the initiation and propagation of cracks are postponed by the strong friction and anchoring force that exists between BF and the substrate. However, the specimen’s fracture growth speed decreases once the peak load and the toughness of the UHPC increase due to the high tensile strength of SF. Results of the SEM scanning study indicate that the hybrid fibers have a multi-level, step-by-step inhibitory effect on cracks, suggesting that UHPC goes through a stable phase of crack development and propagation before failing. At different structural levels and loading stages, the combination of the two results in a superimposition effect and synergistic impact, enabling the fibers to effectively fulfill their individual roles. This achieves synergistic complementarity and the desired outcomes of progressive crack resistance and enhanced toughness.

## 4. Conclusions

This study employed a novel environmentally friendly resin-twisted basalt fiber and steel fiber blend in UHPC, elucidating the impact of individual fiber doping on the mechanical properties of UHPC. Moreover, it compared the differences in the influence of the two types of fibers on the fundamental properties of UHPC. Next, resin-twisted basalt fibers with replacement rates of 25%, 50%, and 75% were used to replace portions of the steel fibers. The mechanical characteristics of UHPC under various replacement rates were examined, the ideal dosage of hybrid fibers in UHPC was estimated, and the mechanism of action of BF and SF in UHPC was investigated. The conclusions that follow are derived from the findings.

(1)The fluidity of UHPC with BF decreased more than that of UHPC with SF. And when the BF content reaches 1.5%, it does not meet the requirements of the fluidity degree. The fluidity of UHPC gradually decreases as the replacement rate of BF rises. The minimum UHPC flow value of 1.5% BF + 0.5% SF is 175 mm, which does not meet the requirements of the flow state UHPC. When the BF replacement rate is below 50%, the fluidity is greater than 200 mm.(2)BF at a dosage of 0–1% improves the flexural strength of UHPC more effectively compared to the SF, and the improvement in early compressive strength is significant. The splitting tensile strength of UHPC first rises and then falls as the fiber content increases. The ideal amounts of BF and SF are 1% and 1.5%, respectively.(3)BF improves the resilience of UHPC, causing a significant rise in the specimen’s initiation load and peak load. Because UHPC-BF has a denser microstructure, the BF and matrix bonding performance is enhanced.(4)As the BF replacement rate rises, the compressive strength of hybrid fiber UHPC progressively falls, while the splitting tensile strength and flexural strength first rise before falling. UHPC made with composite fibers is more durable than that of single fiber. Analysis of the microstructure morphology indicated that the two types of fibers may exert their respective roles at various loading stages, complementing each other in a synergistic manner to execute the effects of progressive fracture resistance and toughening enhancement.(5)The impact resistance of composite fiber UHPC first rises and then falls as the rate of BF substitution rises, and the ductility gradually declines. The initial crack and final failure hammer blows reach their peak values at a dosage of 0.5% BF + 1.5% SF, which is 16.1% and 3.9% higher than that of 2% SF, respectively. The shrinkage rate of UHPC tends to drop initially and then increases with the rise of the BF substitution rate.

## Figures and Tables

**Figure 1 materials-17-03299-f001:**
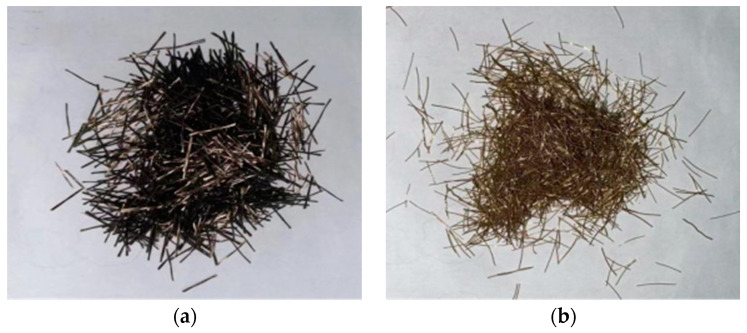
Fibers used in the test. (**a**) BF; (**b**) SF.

**Figure 2 materials-17-03299-f002:**
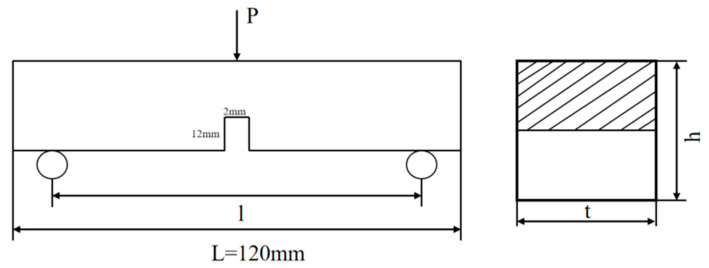
Geometric schematic diagram of the specimen.

**Figure 3 materials-17-03299-f003:**
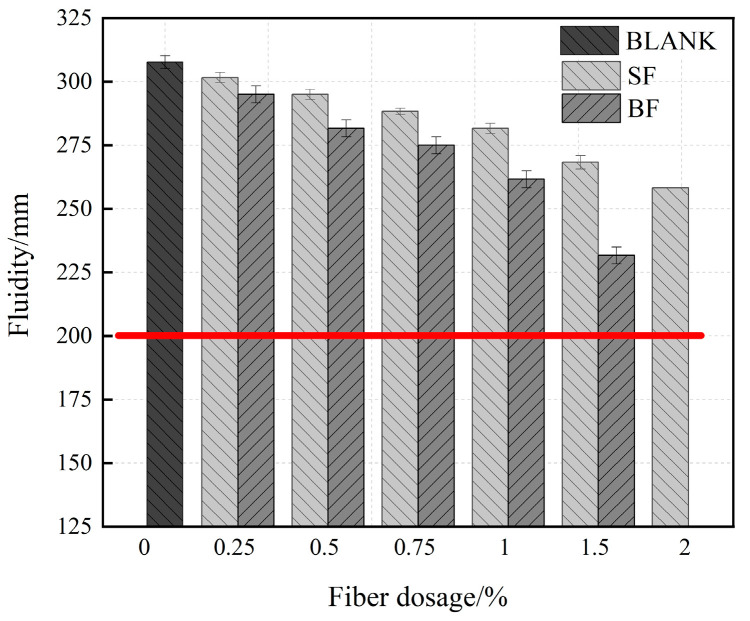
Fluidity of UHPC in single-doped fiber.

**Figure 4 materials-17-03299-f004:**
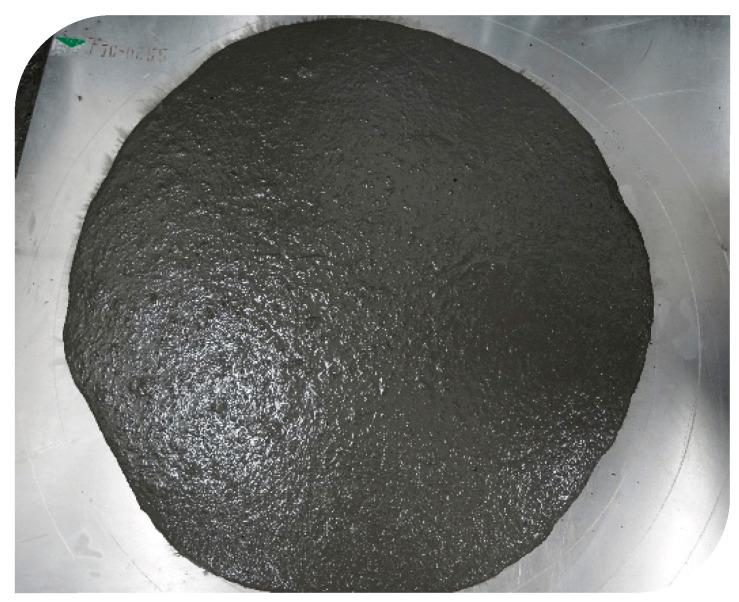
UHPC fluidity test chart.

**Figure 5 materials-17-03299-f005:**
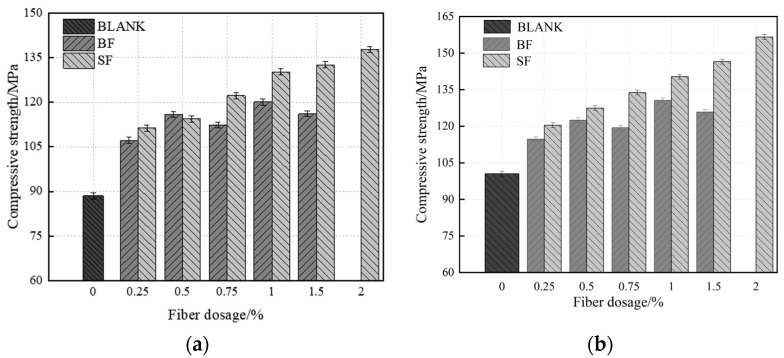
Compressive strength of UHPC in single-doped fiber. (**a**) 7 d; (**b**) 28 d.

**Figure 6 materials-17-03299-f006:**
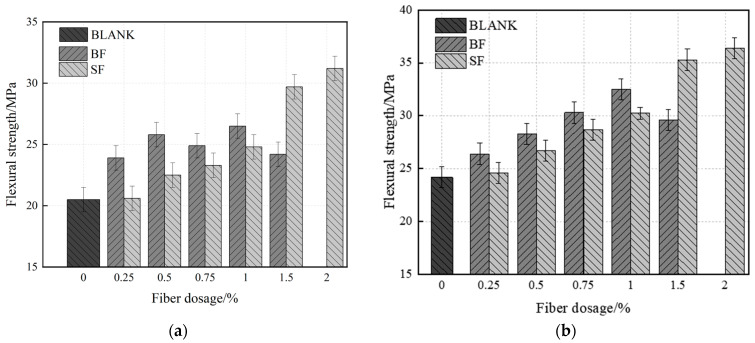
Flexural strength of UHPC in single-doped fiber. (**a**) 7 d; (**b**) 28 d.

**Figure 7 materials-17-03299-f007:**
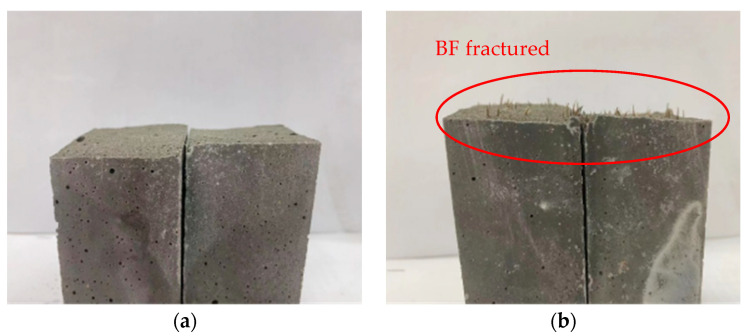
Failure morphology of UHPC specimens. (**a**) Blank UHPC; (**b**) UHPC-BF.

**Figure 8 materials-17-03299-f008:**
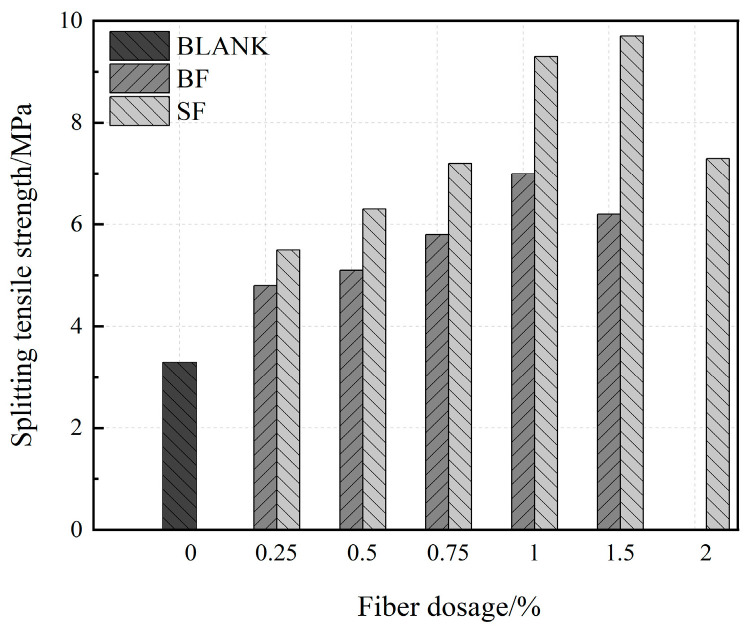
Splitting tensile strength of UHPC in single-doped fiber at 28 days.

**Figure 9 materials-17-03299-f009:**
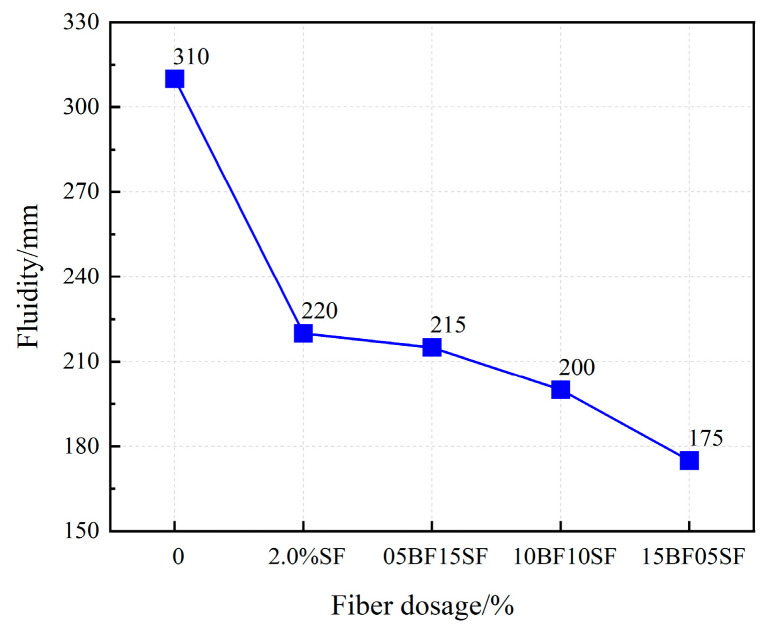
Fluidity of UHPC in hybrid fibers.

**Figure 10 materials-17-03299-f010:**
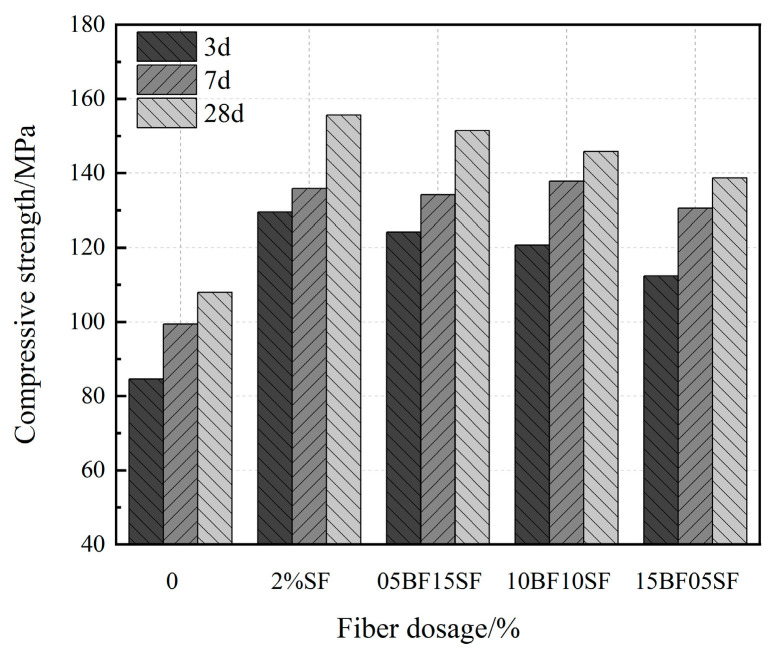
Compressive strength of UHPC in hybrid fibers.

**Figure 11 materials-17-03299-f011:**
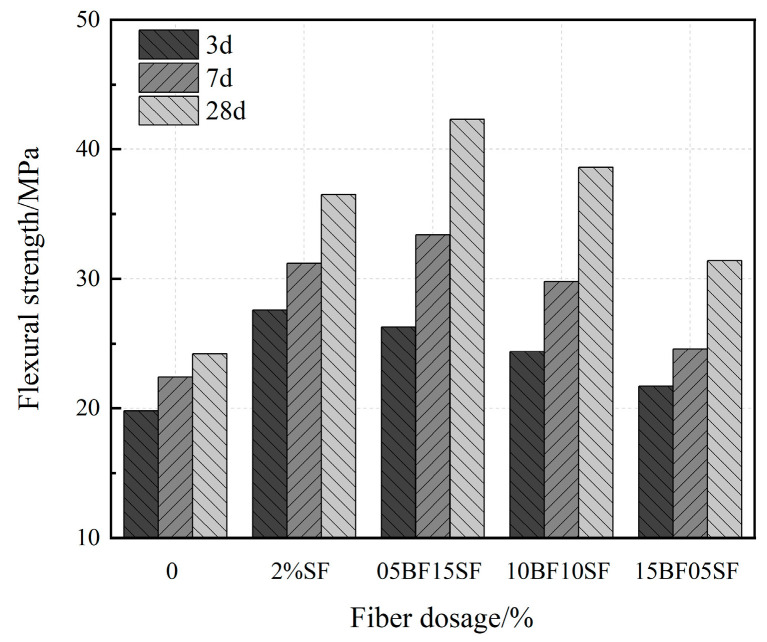
Flexural strength of UHPC in hybrid fibers.

**Figure 12 materials-17-03299-f012:**
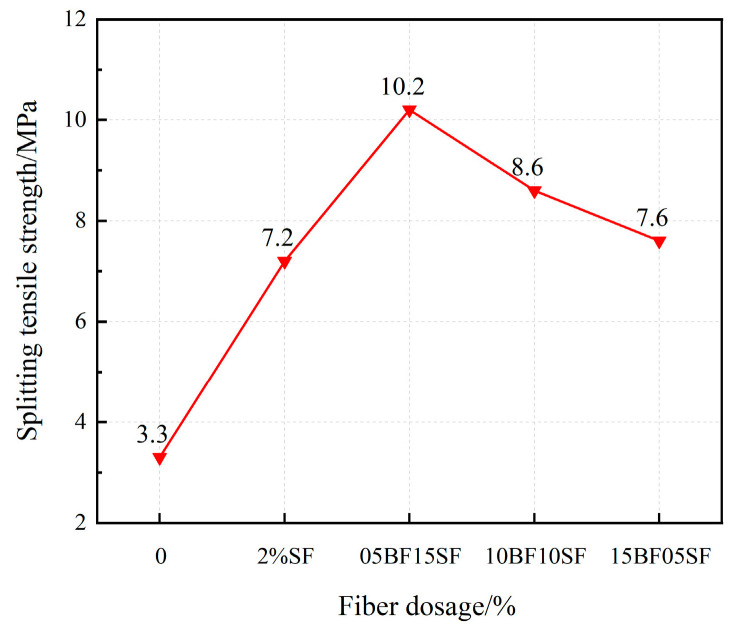
Splitting tensile strength of UHPC in hybrid fibers.

**Figure 13 materials-17-03299-f013:**
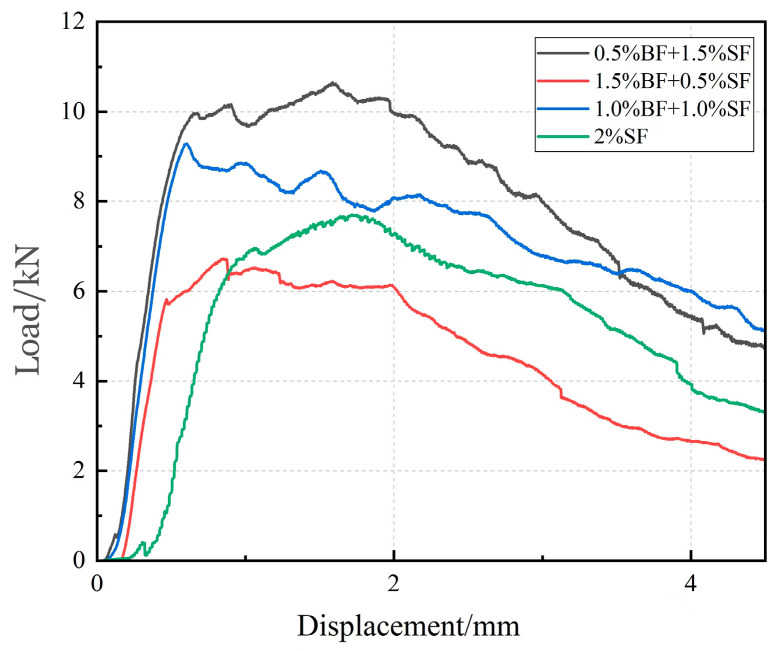
Load deformation curve of specimens.

**Figure 14 materials-17-03299-f014:**
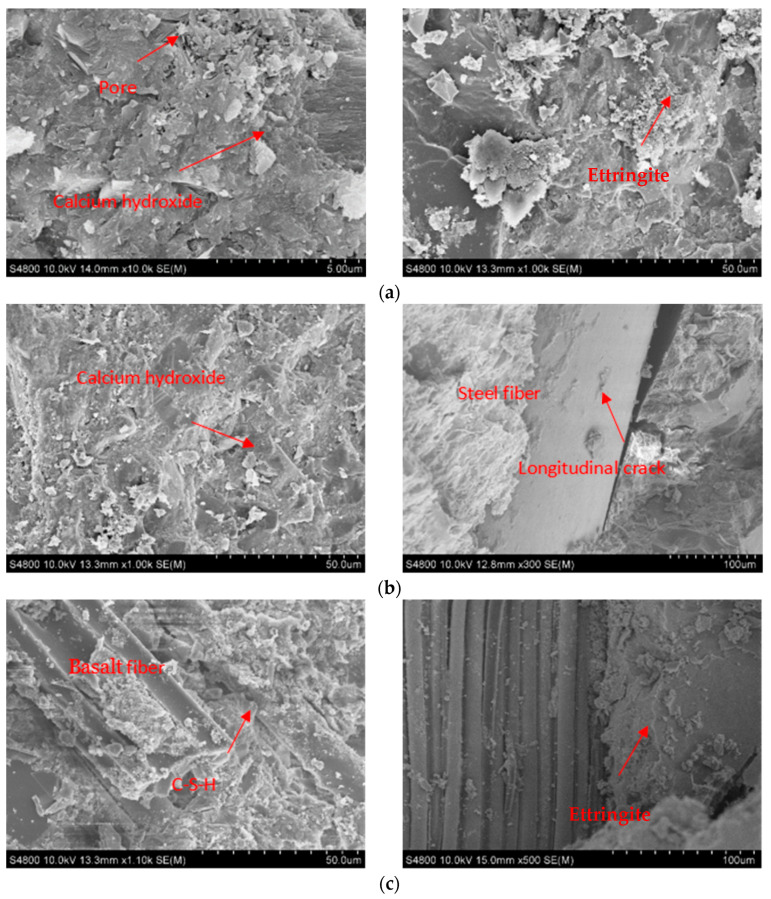
SEM microscopic morphology diagram. (**a**) Blank UHPC at 28 d; (**b**) UHPC-SF at 28 d; (**c**) UHPC-BF at 28 d.

**Figure 15 materials-17-03299-f015:**
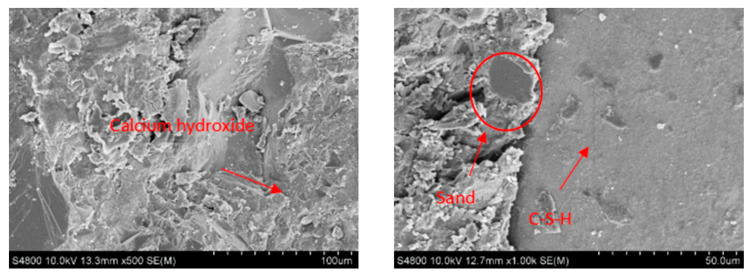
SEM image of 05BF15SF at 28 d.

**Table 1 materials-17-03299-t001:** Fiber-characteristic parameters.

Fiber Type	Diameter/μm	Length/mm	Tensile Strength/MPa	Elastic Modulus	Elongation at Break/%
				/GPa	
BF	450	12	1245	52.3	2.2
SF	300	12	1200	210	3.5

**Table 2 materials-17-03299-t002:** Mix proportion design of fiber UHPC.

Specimen	BF/kg	SF/kg	Cement/kg	Silica Fume/kg	Slag/kg	Sand/kg	Water Reducer/kg	Water/kg
BLANK	0	0	1120	145	105	1120	21	228
0.25% BF	6.3	0	1120	145	105	1120	21	228
0.5% BF	12.7	0	1120	145	105	1120	21	228
0.75% BF	19.0	0	1120	145	105	1120	21	228
1% BF	25.4	0	1120	145	105	1120	21	228
1.5% BF	38.1	0	1120	145	105	1120	21	228
0.25% SF	0	20.8	1120	145	105	1120	21	228
0.5% SF	0	41.6	1120	145	105	1120	21	228
0.75% SF	0	62.4	1120	145	105	1120	21	228
1% SF	0	83.3	1120	145	105	1120	21	228
1.5% SF	0	124.9	1120	145	105	1120	21	228
2% SF	0	166.6	1120	145	105	1120	21	228
05BF15SF	12.7	124.9	1120	145	105	1120	21	228
10BF10SF	25.4	83.3	1120	145	105	1120	21	228
15BF05SF	38.1	41.6	1120	145	105	1120	21	228

**Table 3 materials-17-03299-t003:** Fracture energy of various types of UHPC.

Fibre Type	Cracking Load	Peak Load	Fracture Energy
	(KN)	(KN)	KJ/m^3^
BLANK	2.12	3.88	0.26
1% BF	4.28	6.45	2.36
1% SF	3.22	5.01	13.51
2% SF	4.378	9.2503	25.8
05BF15SF	6.783	9.81893	27.9
10BF10SF	5.965	8.37683	24.3
15BF05SF	3.886	5.94573	17.5

## Data Availability

The original contributions presented in the study are included in the article, further inquiries can be directed to the corresponding author.

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
