# Peer review of "Effect of Basalt/Steel Individual and Hybrid Fiber on Mechanical Properties and Microstructure of UHPC"

_materials, 2024, doi:10.3390/ma17133299_

Round 1
Reviewer 1 Report
Comments and Suggestions for Authors
Fiber reinforcement is widely used to improve the performance of concrete under various types of service loads. Among the various types of fibers, hybrid fibers have proven to be more effective, apparently due to the combination of mechanical properties of different types of fibers. Overall, the manuscript submitted for review represents a comprehensive study of the effects of using a novel resin-spun basalt fiber and steel fiber blend in UHPC, providing valuable information on the optimal fiber composition and the underlying mechanisms determining material performance. The results of the conducted research may have practical application.
There are several comments and questions regarding the text of the manuscript.
1. It is necessary to indicate how the mechanical characteristics of basalt fiber, which are presented in Table 1, were determined. If you used ready-made results, then add a link to the source. In particular, in work [1A]: you can find slightly different mechanical properties of basalt fibers (for example, "...the tensile strength is between 2600 to 4840 MPa, and the range of the elastic modulus is 80–115 GPa").
1A. Bheel, N. Basalt fibre-reinforced concrete: review of fresh and mechanical properties. J Build Rehabil 6, 12 (2021). https://doi.org/10.1007/s41024-021-00107-4
2. In the subsection "2.2. Mix designs", the authors have well described the formulation of concrete mixes. However, it is necessary to indicate the peculiarities of the technology of making mixtures, in particular, at what stage the fiber was introduced, if basalt and steel fiber were used at the same time, then in what sequence they were introduced, with the help of which devices the mixture was made. It is necessary to communicate all the nuances so that the reader can fully reproduce the process of your experiment.
3. Apparently, in line 170, the symbol "DELTA ZERO" was mistakenly omitted before the phrase "is the displacement of the loading point during final failure (mm).
4. How do the findings on the ideal dosage of hybrid fibers in UHPC compare to the existing literature on fiber-reinforced UHPC? Are the proposed optimal fiber contents supported by previous research or do they present new insights?
5. Intuitively, the bond strength between the fibre and the concrete matrix is ​​the dominant mechanism influencing the degree of improvement of concrete due to fibre reinforcement. Why do you use a fibre length of 12 mm (see Table 1)? Should it be varied depending on the dimensions of the building product?
6. What are the potential practical applications of the research findings? Are the improvements in mechanical properties and durability demonstrated in this study significant enough to justify further development and implementation?
Author Response
I would like to thank you for taking the time and effort to go through my paper and providing constructive criticisms which are extremely valuable for me. I appreciate your thoughtful review and am grateful for your valuable insights. I believe those changes have significantly improved the paper and I thank you again for your careful review. I have made all the suggested changes, and I hope my paper meets your approval.
Comments 1: A. Bheel, N. Basalt fibre-reinforced concrete: review of fresh and mechanical properties. J Build Rehabil 6, 12 (2021).
https://doi.org/10.1007/s41024-021-00107-4.
Response 1: Thank you for the suggestion. Basalt fiber we used was provided by the enterprise, and its mechanical properties were written according to the data report provided by the enterprise, so it was a specific value.
Comments 2: In the subsection "2.2. Mix designs", the authors have well described the formulation of concrete mixes. However, it is necessary to indicate the peculiarities of the technology of making mixtures, in particular, at what stage the fiber was introduced, if basalt and steel fiber were used at the same time, then in what sequence they were introduced, with the help of which devices the mixture was made. It is necessary to communicate all the nuances so that the reader can fully reproduce the process of your experiment.
Response 2: Thank you for the suggestion, we have provided additional explanations in the subsection "2.2", as shown below.
The UHPC specimens were prepared by JJ-5 cement mortar mixer. Cementitious material (cement, mineral admixtures) and sand were put into the mixing pot and stirred for 2 mins to ensure good dispersion and achieve higher fiber matrix interface performance. The superplasticizer and water were mixed evenly, then half of them were added into the mixing pot and stirred for 6 mins. The fibers were added to the mixtures lowly, and stirred for 2mins to disperse evenly. Finally, the UHPC slurry was poured into the mold, and the specimen was formed by shaking table for 2mins. The formed specimen was put into the standard maintenance room for 24 hours, and then removed into the standard curing room until the specified age.
Comments 3: In line 170, the symbol "DELTA ZERO" was mistakenly omitted before the phrase "is the displacement of the loading point during final failure (mm).
Response 3: Thank you for the suggestion, we have added the symbol “” before the phrase "is the displacement of the loading point during final failure (mm)”.
Comments 4: How do the findings on the ideal dosage of hybrid fibers in UHPC compare to the existing literature on fiber-reinforced UHPC? Are the proposed optimal fiber contents supported by previous research or do they present new insights?
Response 4: Thank you for the suggestion, we have provided additional explanations, as shown below.
In this paper, SF was partially replaced by BF with the replacement rates of 25%, 50% and 75%, and the ideal dosage of mixed fiber in UHPC was determined by studying the working and mechanical properties of hybrid fiber UHPC. Previous studies have shown that mixing mixed fibers into UHPC can effectively improve its mechanical properties, durability and crack resistance. New insights may lie in the optimal combination of different types and proportions of fibers, such as steel fiber and resin-stranded basalt fiber in this paper, to obtain the best performance in a specific engineering application.
Comments 5: Intuitively, the bond strength between the fibre and the concrete matrix is the dominant mechanism influencing the degree of improvement of concrete due to fibre reinforcement. Why do you use a fibre length of 12 mm (see Table 1)? Should it be varied depending on the dimensions of the building product?
Response 5: Thank you for the suggestion. The previous study "The effect of basalt fiber type and content on the properties of cement mortar and concrete" found that 12mm fiber had the most obvious improvement on the properties of concrete. Therefore, 12mm steel fiber and basalt fiber were selected to be incorporated into UHPC for research.
Comments 6: What are the potential practical applications of the research findings? Are the improvements in mechanical properties and durability demonstrated in this study significant enough to justify further development and implementation?
Response 6: Thank you for the suggestion, we have provided additional explanations, as shown below.
Hybrid fiber reinforced ultra-high performance concrete (UHPC) can be used in construction, railway and bridge engineering. It is used to repair and reinforce the ground structure of roads and airports, providing excellent wear resistance and load bearing capacity while reducing the formation and spread of cracks. Hybrid fiber UHPC generally shows higher strength and toughness than conventional concrete, and this performance improvement makes UHPC a distinct advantage in structures that need to withstand high strength and harsh environmental conditions. It can also significantly reduce the formation and expansion of cracks, thereby improving the durability and service life of the structure, which is of great significance for reducing maintenance costs and extending the life of the structure.
Reviewer 2 Report
Comments and Suggestions for Authors
The manuscript “Effect of Basalt/Steel Individual and Hybrid Fiber on Mechanical Properties and Microstructure of UHPC” contains inaccuracies and understatements. The authors did not clearly present how the research was conducted and how many samples each test was performed on, taking into account the standard deviation. This needs clarification and correction in the article.
General remarks
1. The article should explain what "high-strength bone material" (34 line) means.
2. In the manuscript, the authors should write what impact the selection of the length of BF and SF fibers had on the strength.
3. The authors should explain why the sample with 2% BF admixture is not shown in Table 1.
4. It should also be explained what the impact was of increasing the weight of SF fibers (2%) to 332.2 kg compared to other UHPC samples.
5. In chapters 2.3.2; - 2.3.4, write down the sizes of the samples and how the tests were carried out.
6. Determine how many samples there were in each test and what the standard deviation (SD) is in subsequent tests (Fig. 3, 5, 6, 8,10, 11).
7. In Eq. 1 the quantities m, δ0 and g are not explained. An explanation should be provided in the article.
Specific remarks
8. 120 line: It should read: Design mixes.
9. 164 line, Fig 2 Mark the dimensions of the sample in the drawing.
10. 347 line, Fig.13: There should be "kN" on the ordinate axis.
I recommend an in-depth review of the manuscript, including comments, to make it an article suitable for publication in the Materials.
In its current state, the article should not be published.
Comments on the Quality of English LanguageMinor editing of English language required.
Author Response
Dear Reviewer,
I would like to thank you for taking the time and effort to go through my paper and providing constructive criticisms which are extremely valuable for me. I appreciate your thoughtful review and am grateful for your valuable insights. I believe those changes have significantly improved the paper and I thank you again for your careful review. I have made all the suggested changes, and I hope my paper meets your approval.
Comments 1: The article should explain what "high-strength bone material" (34 line) means.
Response 1: Thank you for the suggestion, we have redescribe the definition of UHPC, as shown below. Ultra-high performance concrete (UHPC) is a novel type of concrete material with distinct performance indices [1,2], comprising cement,mineral admixtures, admixtures, fine aggregates, fibers, etc.
Comments 2: In the manuscript, the authors should write what impact the selection of the length of BF and SF fibers had on the strength.
Response 2: Thank you for the suggestion. The previous study "The effect of basalt fiber type and content on the properties of cement mortar and concrete" found that 12mm fiber had the most obvious improvement on the properties of concrete.(Ref. 38) Therefore, 12mm steel fiber and basalt fiber were selected to be incorporated into UHPC for research.
Comments 3: The authors should explain why the sample with 2% BF admixture is not shown in Table 1.
Response 3: Thank you for the suggestion, we have provided additional explanations, as shown below. When the BF content reached 1.5%, the fluidity of UHPC-BF became lower than 200mm, which did not meet the requirements of fluidity degree, therefore the addition of UHPC-BF was halted. Therefore, we are more inclined to explore these common combinations than the case of using 2% BF alone.
Comments 4: It should also be explained what the impact was of increasing the weight of SF fibers (2%) to 332.2 kg compared to other UHPC samples.
Response 4: Thank you for the suggestion. During the writing process, we mistakenly identified the weight of SF fiber (2%) as 166.6 kg, but this error has been rectified.
Comments 5: In chapters 2.3.2; - 2.3.4, write down the sizes of the samples and how the tests were carried out.
Response 5: Thank you for the suggestion, we have provided additional explanations on the size of the samples and how to conduct testing in chapters 2.3.2-2.3.4, as shown in chapters 2.3.2-2.3.4.
Comments 6: Determine how many samples there were in each test and what the standard deviation (SD) is in subsequent tests (Fig. 3, 5, 6, 8,10, 11).
Response 6: Thank you for the suggestion, we made three specimens for each test, and we have revised the statement and added error bars in the corresponding figures, as shown in Figure 3, Figure 5 and 6. Due to the error being within 10%, no error bars were added to Figures 8, 10, and 11.
Comments 7: In Eq. 1 the quantities m, δ0 and g are not explained. An explanation should be provided in the article.
Response 7: Thank you for the suggestion, we have provided additional explanations for Equation 1, as shown below.
Where G is the fracture energy (J); W is the measured P-δ surrounding area of the curve and abscissa, A is the area of the fractured ligament of the specimen (mm2); is the displacement of the loading point during final failure (mm); m is the mass of the specimen (kg); g is the acceleration of gravity.
Comments 8: 120 line: It should read: Design mixes.
Response 8: Thank you for the suggestion, we have We have replaced "Mix designs" in Line 120 with " Design mixes ". As shown in Line 120.
Comments 9: 164 line, Fig 2 Mark the dimensions of the sample in the drawing.
Response 9: Thank you for the suggestion, we have We have marked the dimensions of the sample in Figure 2.
Comments 10: 347 line, Fig.13: There should be "kN" on the ordinate axis.
Response 10: Thank you for the suggestion, we have corrected the ordinate axis of Figure 13 to "kN".
Round 2
Reviewer 2 Report
Comments and Suggestions for Authors
The authors took into account the comments and comments regarding the manuscript. The article may be published after minor corrections.
Comments on the Quality of English LanguageMinor editing of English language required.